# Experimental Methods of Investigating Airborne Indoor Virus-Transmissions Adapted to Several Ventilation Measures

**DOI:** 10.3390/ijerph191811300

**Published:** 2022-09-08

**Authors:** Lukas Siebler, Maurizio Calandri, Torben Rathje, Konstantinos Stergiaropoulos

**Affiliations:** Institute for Building Energetics, Thermotechnology and Energy Storage (IGTE), University of Stuttgart, Pfaffenwaldring 35, 70569 Stuttgart, Germany

**Keywords:** aerosol infection risk, SARS-CoV-2 transmission, measurement methods, surrogate particles, trace gas, ventilation measures

## Abstract

This study introduces a principle that unifies two experimental methods for evaluating airborne indoor virus-transmissions adapted to several ventilation measures. A first-time comparison of mechanical/natural ventilation and air purification with regard to infection risks is enabled. Effortful computational fluid dynamics demand detailed boundary conditions for accurate calculations of indoor airflows, which are often unknown. Hence, a suitable, simple and generalized experimental set up for identifying the spatial and temporal infection risk for different ventilation measures is more qualified even with unknown boundary conditions. A trace gas method is suitable for mechanical and natural ventilation with outdoor air exchange. For an accurate assessment of air purifiers based on filtration, a surrogate particle method is appropriate. The release of a controlled rate of either trace gas or particles simulates an infectious person releasing virus material. Surrounding substance concentration measurements identify the neighborhood exposure. One key aspect of the study is to prove that the requirement of concordant results of both methods is fulfilled. This is the only way to ensure that the comparison of different ventilation measures described above is reliable. Two examples (a two-person office and a classroom) show how practical both methods are and how the principle is applicable for different types and sizes of rooms.

## 1. Introduction

The ongoing pandemic teaches us that the implementation of appropriate ventilation measures can significantly reduce indoor infection risks [1,2,3,4].

For fast and simple assessments, many calculation models regarding the infection risk of SARS-CoV-2 in indoor environments exist [5,6,7,8,9,10]. Often, these are based on idealised assumptions (e.g., ideal mixed ventilation). In reality, the condition of ideal mixed ventilation is impossible to achieve (finite velocity for distributing locally released substances), especially for larger rooms. Even for small to medium scales, the calculation models may not resolve aerosol dispersion appropriately.

In practice, different ventilation principles—natural ventilation and mechanical ventilation (mixing ventilation, displacement ventilation, downward ventilation)—are applied, depending on the type of use and the occupancy density of the room [11,12]. Depending on the meteorological boundary conditions such as wind velocity, natural ventilation can be classified between mixed ventilation and displacement ventilation [13]. These transient effects cannot be represented by a simple calculation model [14]. Furthermore, existing disturbance influences (leaks and opening of windows/doors, downdrafts, etc.) have to be considered for the respective ventilation type.

Displacement ventilation concepts, often applied in large halls, impede estimations of virus transmission to neighbors. In ideal theory, there would occur only unobjectionable vertical buoyancy flows. In reality, however, these are superimposed by disturbance effects [15,16,17], which might be critical regarding neighbor infection risks. If the relevant boundary conditions for these disturbing influences are unknown, the estimation is regarded to be even more critical.

For individual considerations and higher accuracies, experimental studies examine the effects of ventilation measures on airborne infections. Some of them deal with a real virus load of the indoor air with actually present infectious persons, without controlled boundary conditions [18]. In Li et al. [19], trace gas measurements and flow simulations for an actual infection scenario are conducted in order to reproduce transmission paths in detail. Transient and absolute considerations of virus loads are omitted in this context. Another research study examines the effects of ventilation measures, but does not take into account the transfer of substitutes to viruses and thus the actual infection risk [20]. As a result, only relative statements on the risk reduction are possible.

In general, ventilation measures can be divided into filtering recirculation methods (e.g., air purification) and ventilation principles with outside air exchange (e.g., natural/mechanical ventilation). However, there are no published studies on the comparative assessment of both measures with regard to the infection risk.

Previous experimental research is not yet suitable for a holistic individual assessment of the infection risk. Therefore, a comprehensive and standardized measurement procedure for any airborne transmitted disease is required. In this paper, two consistent measurement methods are presented, which allow for determining spatially and temporally resolved infection risks in rooms accurately. For the first time, the two consistent approaches can be used to evaluate both filtering recirculating air operations and ventilation principles with outdoor air exchange.

## 2. Theory of Airborne Virus Infections

In 1955, the first approach of assessing general infection risks was developed, which was improved and specified in 1978 on the disease measles [21,22]. The authors introduced the dose of inhaled quanta $D_{q}$ as an indicator of whether an infection occurs. Taking into account the efficiency of masks, it can be calculated as:(1)$$\begin{array}{c} D_{q}\left(t\right)={\dot{V}}_{\mathrm{inh}}\;\left(1-\eta_{\mathrm{inh}}\right)\;\int_{0}^{t}c_{q}\left(t\right)\;dt+D_{q\;0} \end{array}$$
with ${\dot{V}}_{\mathrm{inh}}$, $\eta_{\mathrm{inh}}$, $c_{q}\left(t\right)$, and $D_{q\;0}$ as inhalation volume flow, mask filtration efficiency for inhaling, quanta concentration at time *t* and the initial value of $D_{q}$, respectively.

The approach of Wells et al. and Riley et al. allows the computation of the predicted infection risk via aerosols (PIRA), labeled as $P_{I}$ [22]:(2)$$\begin{array}{c} P_{I}=1-e^{-D_{q}}. \end{array}$$

As an alternative, the dose–response model follows a modified principle that is similarly referenced in science and should be roughly presented. Hereby, the number of pathogens that result in infections of a certain proportion of a group is usually determined on the basis of empirical animal experiments. Based on an analysis of how many pathogens a person exhales, the infection risk can be derived. The dose–response as well as a Wells–Riley model are accepted as valuable tools in epidemiological studies. When using these models, the respective advantages, disadvantages and uncertainties must be weighed [23].

In this paper, the Wells–Riley model is followed. However, the relevant equations can be easily modified to a dose–response model, due to substituting quanta emission rates (QER) with pathogen emission rates and an adaption of the risk assessment.

In general, infection risks can be reduced by either ventilation measures (mechanical and natural ventilation) or air purification concepts (without outside air exchange) can be used. For a non-spatially resolved estimate of the quanta concentration, the general differential Equation (Equation 3), which includes various ventilation concepts, has to be evaluated:(3)dcq(t)dt=q˙out(1−ηexh)Vr︸quantareleaseofaninfectiousperson−V˙dev/wcq(t)ηdevVr︸quantaremovalofventilation−Φ˙cq(t)1Vr︸naturalin-activation/deposition,cq(0)=cq0
with ${\dot{q}}_{\mathrm{out}}$, $\eta_{\mathrm{exh}}$, $V_{r}$, ${\dot{V}}_{\mathrm{dev}/w}$, $\eta_{\mathrm{dev}}$, $\dot{\Phi }$ as quanta rate (output), mask filtration efficiency for exhaling, room volume, volume flow of device or window and device efficiency (filtration ratio for air purifiers, exhaust air to outdoor air exchange for ventilation systems, equals 0 for natural ventilation) and combined rate of natural inactivation and deposition of existing viruses, respectively. The last equation term can be described and modelled well in theory, but, in practice, it is challenging to include it in experimental settings. Integrating Equation (Equation 3) over time followed by using (Equation 1) and (Equation 2), the dose of inhaled quanta and finally PIRA can be calculated.

With these assumptions, it is possible to estimate infection risks using ventilation devices, air purifiers and natural ventilation under ideal mixed ventilation conditions. However, experimental methods of substance dispersion concerning the airborne transmission of virus infections enable accurate spatially and temporally resolved results for any ventilation principle even for unknown boundary conditions.

## 3. Experimental Methods

Airborne virus transmission is mostly dealing with particles below 10 μm of size [24,25]. These particles are meant to have negligible sink velocities and follow airflows almost exactly [26]. A common method investigating airflows quantitatively is releasing substances (which also follow the airflow) and measuring their concentrations. In order to evaluate the totality of the ventilation measures, suitable substances are trace gas and surrogate particles. For scenarios with outside air exchange, only trace gas is appropriate because of entering particles from the outside, which would falsify particle measurements. For recirculation devices based on filtration, however, trace gas is inapplicable since it is not filtered and therefore no particle removal can be measured. For this reason, two different methods are essential.

### 3.1. Trace Gas Method

The focus of previous indoor air investigations using trace gas, also conducted at the University of Stuttgart, has mostly been on evaluating the ventilation effectiveness rather than infection risks [27]. The method is based on the emission of trace gas, which is not present in the natural surrounding and whose concentration can be measured by infrared spectrometers (e.g., N_2_O or SF_6_). First, the room needs to be freed from trace gas from eventual previous measurements. By using a mass flow controller with gas-specific adapted properties, a constant and continuous mass flow of the trace gas is possible.

Along with the measured trace gas concentration, the quanta rate (output) and the respiratory rate following theoretical post processing lead to a calculation of the infection risk of a certain disease.

For trace gas, which is transported in a similar way to airborne particles and viruses (below 5 μm) [28], the following assumption without mask filtration efficiency is applied:(4)$$\begin{array}{c} \frac{{\dot{n}}_{\mathrm{in}\;\mathrm{tg}}\left(t\right)}{{\dot{n}}_{\mathrm{out}\;\mathrm{tg}}}=\frac{{\dot{q}}_{\mathrm{in}}\left(t\right)}{{\dot{q}}_{\mathrm{out}}} \end{array}$$
with ${\dot{n}}_{\mathrm{in}\;\mathrm{tg}}$, ${\dot{n}}_{\mathrm{out}\;\mathrm{tg}}$, ${\dot{q}}_{\mathrm{in}}$, ${\dot{q}}_{\mathrm{out}}$ as (fictitious) molar flow for trace gas (input), molar flow for trace gas (output), quanta rate (input) and quanta rate (output) respectively. Note that input and output represent a fictitious inhalation and exhalation.

From the (time dependent) quanta rate (input) ${\dot{q}}_{\mathrm{in}}$, the dose of inhaled quanta $D_{q}$ is determined by integration over time (assumption: $D_{q\;0}=0,\;c_{q\;0}=0$):(5)$$\begin{array}{c} D_{q}=\int_{0}^{t}{\dot{q}}_{\mathrm{in}}\left(t\right)\;dt=\frac{{\dot{q}}_{\mathrm{out}}}{{\dot{n}}_{\mathrm{out}\;\mathrm{tg}}}\;\int_{0}^{t}{\dot{n}}_{\mathrm{in}\;\mathrm{tg}}\left(t\right)\;dt. \end{array}$$

The molar flow for (fictitious) trace gas input is calculated by using $\dot{n}=\dot{m}/M$ and the assumption $c_{\mathrm{tg}}\left(t\right)\ll 1\;{\mathrm{mol}}_{\mathrm{tg}}/{\mathrm{mol}}_{\mathrm{air}}$:(6)$$\begin{array}{c} {\dot{n}}_{\mathrm{in}\;\mathrm{tg}}\left(t\right)=c_{\mathrm{tg}}\left(t\right)\;{\dot{n}}_{\mathrm{inh}}=c_{\mathrm{tg}}\left(t\right)\;\frac{\rho_{\mathrm{air}}\;{\dot{V}}_{\mathrm{inh}}}{M_{\mathrm{air}}} \end{array}$$
with $c_{\mathrm{tg}}$, ${\dot{n}}_{\mathrm{inh}}$, $\rho_{\mathrm{air}}$, and $M_{\mathrm{air}}$ as measured trace gas concentration, inhalation molar flow, density and molar mass of air, respectively.

The resulting equation for the dose of inhaled quanta $D_{q}$, considering mask filtration efficiency (see [29,30]), is:(7)$$\begin{array}{cc} D_{q} & =\int_{0}^{t}{\dot{q}}_{\mathrm{in}}\left(t\right)\;dt\\ \; & =\left(1-\eta_{\mathrm{inh}}\right)\;\left(1-\eta_{\mathrm{exh}}\right)\;\frac{{\dot{q}}_{\mathrm{out}}\;M_{\mathrm{tg}}}{{\dot{m}}_{\mathrm{out}\;\mathrm{tg}}}\;\frac{\rho_{\mathrm{air}}\;{\dot{V}}_{\mathrm{inh}}}{M_{\mathrm{air}}}\;\int_{0}^{t}c_{\mathrm{tg}}\left(t\right)\;dt \end{array}$$
with $M_{\mathrm{tg}}$, ${\dot{m}}_{\mathrm{out}\;\mathrm{tg}}$ as molar mass and mass flow of trace gas (output), respectively.

A numerical integration of the concentration of trace gas over time $c_{\mathrm{tg}}\left(t\right)$ (e.g., via trapezoidal rule) allows the transfer of measurement data via Equation (Equation 2) into an infection risk for scenarios with outdoor air exchange. Furthermore, data for the quanta rate (output) are required, which can be obtained from literature or identified by a backward processing of this introduced approach, (see Section 6).

### 3.2. Surrogate Particles Method

To evaluate the function of an air purifier based on filtration, an alternative method is required because trace gas can not be filtered. Even though the approaches have analogous equations, they still involve different physical units and more elaborate conditioning, since the room needs to be initially particle-free. From then on, only particles that are actually released by particle generators are measured. In reality, it cannot be avoided that a certain number of particles is still present. Superimposing this concentration with high emissions is a reasonable measure. If the particle emission rate is set very high, particles coming from outside (e.g., via infiltrations) have no influence on the measurement results because their particle rate is several powers of ten lower. In this way, a simple and cost-efficient measuring technique is enabled, which can be widely applied.

A suitable substance for particles is Di-Ethylhexyl-Sebacat (DEHS). Besides the advantage that its particles keep the same size over time because of very low evaporation rates, the measured size distribution is similar to the exhaled particles of humans [25,31].

Any other possible particle sources (even persons) should be removed from the room of investigation or kept emitting as low as possible. Using several optical particle counters (OPC) or scanning mobility particle sizers (SMPS) on different positions allow a spatial and temporal measurement of the number concentration and size distribution of particles.

The measured surrogate particle concentration, the quanta rate (input) and the respiratory rate lead to the calculation of the infection risk for a room equipped with an air purifier. Furthermore, data for the quanta rate (output) are required, which can be obtained from literature.

Analogous to the trace gas method for surrogate particles, the following assumption without mask filtration efficiency is applied:(8)$$\begin{array}{c} \frac{{\dot{m}}_{\mathrm{in}\;\mathrm{sp}}\left(t\right)}{{\dot{m}}_{\mathrm{out}\;\mathrm{sp}}}=\frac{{\dot{q}}_{\mathrm{in}}\left(t\right)}{{\dot{q}}_{\mathrm{out}}} \end{array}$$
with ${\dot{m}}_{\mathrm{in}\;\mathrm{sp}}$, ${\dot{m}}_{\mathrm{out}\;\mathrm{sp}}$ as mass flow for surrogate particles input and output, respectively (fictitious inhalation and exhalation of particles).

Similar to the trace gas method, the dose of inhaled quanta $D_{q}$ is (assumption: $D_{q\;0}=0,\;c_{q\;0}=0$):(9)$$\begin{array}{c} D_{q}=\int_{0}^{t}{\dot{q}}_{\mathrm{in}}\left(t\right)\;dt=\frac{{\dot{q}}_{\mathrm{out}}}{{\dot{m}}_{\mathrm{out}\;\mathrm{sp}}}\;\int_{0}^{t}{\dot{m}}_{\mathrm{in}\;\mathrm{sp}}\left(t\right)\;dt. \end{array}$$

Substituting the assumed constant mass flow (output) of surrogate particles (${\dot{m}}_{\mathrm{out}\;\mathrm{sp}}=c_{\mathrm{out}\;\mathrm{sp}}\;{\dot{V}}_{\mathrm{ag}}$) and a fictitious mass inhalation rate of surrogate particles (${\dot{m}}_{\mathrm{in}\;\mathrm{sp}}\left(t\right)=c_{\mathrm{in}\;\mathrm{sp}}\left(t\right)\;{\dot{V}}_{\mathrm{inh}}$) in (Equation 9) (considering mask filtration efficiency see [29,30]) results in:(10)$$\begin{array}{c} D_{q}=\int_{0}^{t}{\dot{q}}_{\mathrm{in}}\left(t\right)\;dt=\left(1-\eta_{\mathrm{inh}}\right)\;\left(1-\eta_{\mathrm{exh}}\right)\;\frac{{\dot{q}}_{\mathrm{out}}}{c_{\mathrm{out}\;\mathrm{sp}}\;{\dot{V}}_{\mathrm{ag}}}\;{\dot{V}}_{\mathrm{inh}}\;\int_{0}^{t}c_{\mathrm{in}\;\mathrm{sp}}\left(t\right)\;dt \end{array}$$
with ${\dot{V}}_{\mathrm{ag}}$, $c_{\mathrm{out}\;\mathrm{sp}}$, $c_{\mathrm{in}\;\mathrm{sp}}\left(t\right)$ as volume flow of aerosol generator and mass concentration of surrogate particles (output and input), respectively.

With regard to suspected agglomeration effects, it seems appropriate to extend the detected bandwidth of the emission size distribution upwards compared to the emission bandwidth and to operate via mass-related units. If particles agglomerate, this has an influence on the particle number concentration but not on the particle mass concentration and the infectivity. Therefore, the mass concentration should be applied (in this case, up to an optical diameter of 10 μm, see also Section 6). This can be determined by cumulating mass concentrations of each size fraction by calculating their volumes and their weight with the density of the specific particle substance.

## 4. Comparison between the Two Methods

One key aspect of the study is to prove that both methods are concordant. Hence, a comparison between the two measuring methods for substance dispersion is essential. Bivolarova et al. [32] compare trace gases with different particle sizes on the basis of steady-state observations and decay curves of substance concentrations. A temporal resolution of these concentrations from the initial to the steady state—i.e., the dispersion of the substances—as well as a transfer to an infectious event is not focused here.

However, depending on the infectiousness of a disease and the exposure time, the transient dispersion is relevant to an infection process. In addition, the relation of substance release rates (emission) to emerging concentration profiles (emission) is not yet examined and thus a transfer to a fictitious unit such as quanta is not possible.

For these reasons, an experimental comparison is conducted by the authors with respect to transient quanta concentration courses in an airflow visualization laboratory with controlled boundary conditions.

In this laboratory, both a controlled supply and exhaust air volume flow is adjusted. At a specific position (see Figure 1), controlled flows of both a DEHS particle-laden aerosol (with known emission particle concentration) and a SF_6_ trace gas are released simultaneously. Data of the aerosol generator for the particle release and the mass flow controller for trace gas are given in Table 1. OPCs and sampling tubes for the FTIR gas analyser are placed at six positions in the laboratory. This allows the concentrations of both the particles and the trace gas to be measured simultaneously. Since the authors have six OPCs but only one FTIR at their disposal, the suction of the six trace gas sampling points is changed automatically every minute. In order to take into account the time required for the gas to pass through these tubes to the FTIR, the measured values of each first 30 s are discarded. Switching the sampling points between six positions results in a 6-fold lower sampling rate compared to the OPCs (see Figure 2). In order to ensure that almost no outdoor particles enter the room, a HEPA 14 filter (corresponds with Minimum Efficiency Reporting Values (MERV) 19) was integrated into the duct of supply air. Otherwise, in addition to the released particles, those from the outside would also be measured and therefore corrupts the experimental data. A simultaneous release and measuring of particles and trace gas at the same position are expected to result in the same dispersion of both substances. In this case, the calculated course of the curves of quanta concentrations for both substances is supposed to be similar in all measuring positions.

In Table 1, the parameters of the measurement devices are shown.

The aerosol is generated with a particle number concentration of $1.5\;\times \;10^{7}\;{cm}^{-3}$, which are approximately logarithmic normal distributed (median optical diameter ≈ 0.3 μm). The OPCs are able to measure particle size distributions of 0.175 … 20 μm over 64 channels. Furthermore, data for the quanta rate (output) are required, which can be obtained from literature, see Peng et al. [33].

The experimental set up for the comparative measurement is presented in Figure 1. For the delta variant of SARS-CoV-2, a medium spreader is assumed by Peng et al. [33]. The corresponding boundary conditions are shown in Table 2.

Deviating from usual recommendations of a certain air exchange rate, reference is given to the personal volume flow, see Kriegel et al. [2]. Depending on the viral load, exposition time, non-pharmaceutical measures, etc., this value may differ. However, this paper will focus on the comparison and concordance of the two measurement methods.

Over the entire measurement period, both quanta concentration courses are similar at all measurement positions. Therefore, it is expected that both methods are suitable and can be applied for different ventilation cases. In Figure 2, the exemplary result at a single measuring position (best fit) in the conference room is shown. In addition, it is worth emphasizing the small discrepancy between the two experimental curves compared to the theoretical one after a quasi-steady state has been reached. This is due to high air exchange rates and the possibility of accurate volume flow measurements (correlation to steady state concentration) in the laboratory. However, during the transient course, a discrepancy is generally observed, since substances in reality need time to disperse to a certain position.

Different measurement techniques can generally lead to different sampling rates. In this measurement, only one infrared spectrometer is used for the trace gas concentrations. By means of a measuring position switch, six positions are taken into account successively over one minute each. On the other hand, six OPCs simultaneously record the particle concentrations of a single measuring position every minute. Therefore, the sampling rate for trace gas is six times lower compared to particles.

## 5. Experimental Studies

In the following section, it is shown that both methods are practical. The measured scenarios go from small to medium scale and contain the two of three main ventilation types: a two-person office (air purifier) and a typical classroom (natural ventilation vs. air purifier). These investigated example scenarios are shown in detail.

Besides supply and extract air effects, indoor airflows are influenced by sub- and over-tempered surfaces (e.g., thermal sources). For an appropriate simulation of human heat outputs for a certain activity rate (low activity: 75W) [34], thermal dummies with a surface area of an average human (${1.8\;m}^{2}$) [35] are recommended. Figure 3 shows a professional thermal person dummy and also a low cost variant (cartons with lightning bulbs inside) for substituting many persons in large rooms (e.g., Stuttgart Drama Theatre).

### 5.1. Spatially Resolved Measurements in a Laboratory

Under reproducible boundary conditions in an air visualisation laboratory, an air purifier has been investigated by the authors. The focus lies on the particle removal effectiveness of the device in a two-person office.

Thereby, one of both people is assumed to be infectious. Figure 4 shows a photo and the top view of the experimental set up; the relevant parameters of the experiments concerning the effectiveness of air purifiers are described in Table 3; for the corresponding parameters of the measurement devices, see Table 1.

The particle measurement is analogous to the procedure described in Section 4. However, the filtered outdoor air is now replaced by recirculated air of a filtering air purifier. First, the room is freed from particles by the air purifier. Thereafter, the aerosol is subsequently emitted from the marked dummy (see Figure 4) with the properties of Table 1 over 60 min. The particle concentrations are determined by the OPCs at six different positions in the room in order to obtain spatial information about the substance removal.

The temporally-resolved data allow a computation of the infection risk (PIRA) at these positions using Equation (Equation 10). An exemplary measurement result is shown in Figure 5.

Depending on the ventilation principles (mixing ventilation, displacement ventilation, downward ventilation), the spatial deviations from the assumption of ideal mixed ventilation (in relation to the entire room) may vary. The deviations of the individual positions among each other show a significant benefit of a spatial view. Moreover, quanta concentrations in Figure 5 do not only mismatch the ideal mixed ventilation in the transient range but also in the steady state. In this case for a duration of 1 h, the overall infection risk (PIRA) for a medium spreader (delta variant) by Peng et al. [33] using Equation (Equation 2) results in a range from 2.2% (position 4) to 4.0% (position 2).

### 5.2. Comparison of Ventilation Measures in Classrooms

Further investigations of ventilation measures in classrooms were conducted by the authors, in order to assess the effectiveness of periodic window ventilation, decentralised ventilation systems and air purifiers regarding infection risk and thermal comfort. For this study, it is required that both methods (trace gas and surrogate particles) perform similarly. The set up and the three ventilation measures are shown in Figure 6 exemplarily.

Table 4 describes the relevant parameters of both experiments in an exemplary classroom, and Table 1 shows the corresponding parameters of the measurement devices. The weather data given are mean values of a meteorological station within the measurement period. The weather data are particularly important for window ventilation. They are irrelevant for the operation of the air purifier, since this experiment was carried out separately.

Due to changing boundary conditions during this study (compared to reproducible conditions in the laboratory), only one exemplary classroom is considered for an air purifier and a periodic window ventilation (in this case, 20/5/20: 20min closed, 5min opened). For an assumed medium-spreader (SARS-CoV-2, delta variant) [33] and an exposition time of 1.5 h in both scenarios, there is the same position of the infectious person (substance release).

The investigation of air purification and natural ventilation is carried out successively. The experimental method for the air purifier is carried out in the same way as described in Section 5.1 using DEHS-particles. The reliable measurement of natural ventilation is conducted with trace gas. In this case, SF_6_ is released in a controlled manner analogous to Section 4, and its concentration is measured at six different positions. The periodic window opening is carried out manually according to a stopwatch. The room geometry and dimensions as well as the detailed set-up of the experiment including the positions of the substance release, the measurement and the air purifier are shown in Figure 7.

At two different measurement positions (1 and 4), the quanta concentrations are illustrated in Figure 8. The varying curves show how valuable both spatial resolved methods are.

Besides the position of an infectious person, the location of exposed persons and the air in-/outlets have a significant influence on the infection risks, especially for a non-continuous ventilation via windows. In this particular case, position 4 is proximate to the window front and therefore experiences higher local ventilation rates, which results in a lower quanta concentration course. Although this position is also close to the air purifier, in this situation, it experiences lower local ventilation rates than position 1 and thus a higher quanta concentration course. It is assumed that this anomaly is based on the occurring indoor airflow due to an upwards directed supply air. Besides the possibility of detailed local analysis, this plot emphasizes that the assumption of a theoretical ideal mixed ventilation is inappropriate in this example.

In scenarios with both principles (filtering device and ventilation with outdoor air exchange), like an air purifier combined with a window or mechanical ventilation, a new challenge arises. Outdoor air contains many particles of the same sizes as released by an aerosol generator, which means that an OPC cannot separate between surrogate and outdoor particles. One pre-study in this project shows that a simultaneous outdoor SMPS measurement of particle size distribution provides a solution. There are usually overlaps of the size distributions of outdoor particles and released ones. Consequently, if only particles outside this overlap are considered for both release and detection, the effect of both ventilation measures can be evaluated.

## 6. Discussion

There are two different models for estimating infection risks. Both Wells–Riley and dose–response models are accepted in the scientific community as valuable tools. Even though both experimental methods are easily adaptable, the formulas used in this paper are based on Wells–Riley. One of its criticisms is the uncertainty of quanta emission rate (QER) determination, using a backward calculation for the assumption of an ideal mixed ventilation [23]. Agreeing on that criticism, it should be suggested to use the two introduced experimental methods as well for an accurate determination of these values. With an experimental reconstruction of several infection scenarios, and a backward procedure to the one introduced in this paper, QER could be determined more reliably. By iteratively adjusting a still fictitious QER, infection risks could be derived for all exposed persons. Taking into account numerous scenarios and the persons actually infected, a mean resilient value could be identified using appropriate stochastic tools. This procedure enables a consideration of any kind of ventilation principle, instead of being limited to an ideal mixed ventilation.

A closer look at the experimental methods reveals that operating with surrogate particles (compared to trace gas) has higher uncertainties due to agglomeration, sedimentation and deposition effects. The intensity of agglomeration effects is discussed on the basis of measurements related to Figure 2, which is illustrated in a more detailed version (particle number concentration, particle mass concentrations PM1/PM2.5/PM10) in Figure 9.

The assessment on how well the quanta curves determined by surrogate particle method fit relatively to the ones by trace gas method is calculated as follows:(11)$$\begin{array}{c} \Psi =\frac{\int_{0}^{t}\left|c_{q\;\mathrm{ref}}\left(t\right)-c_{q}\left(t\right)\right|\;\mathrm{dt}}{\int_{0}^{t}c_{q\;\mathrm{ref}}\left(t\right)\;\mathrm{dt}} \end{array}$$
with Ψ and $c_{q\;\mathrm{ref}}$ as curve agreement evaluation and reference quanta concentration (trace gas method), respectively. Table 5 shows the results of the evaluation on the curve agreement between the various approaches.

Since particle number concentration decreases due to agglomeration effects, this results in Ψ=48.7%. For mass concentrations considered below 1 μm, agglomeration might cause particles leaving the upper limit of the OPC’s detection range, which results in lower calculated quanta concentrations compared to trace gas. For 2.5 μm, this effect becomes smaller because of a lower amount of large particles. It is even negligible for 10 μm, and its curve almost matches the trace gas course (Ψ=5.6%). To avoid underestimated infection risks, it is highly recommended to consider mass concentrations up to 10 μm even though the median size of particle release is below 1 μm. Several studies suggest that infectious particles, which remain suspended in the air, can be much larger [36] However, the release of particles with a median of less than 1 µm provides an overestimation of the infection risk. Apart from the lower filter removal efficiency for these particle sizes, they are also more likely to be airborne, resulting in fewer deposition effects. Even if in reality larger particles are emitted by humans, these two aspects ensure a conservative assessment of the infection processes.

Particle agglomeration further impacting both methods might be caused by wall effects, whereas trace gas should be reflected, and liquid particles are assumed to be trapped at walls. This might be one possible explanation for a lower curve of particle mass compared to trace gas concentration (Figure 9).

## 7. Conclusions

In order to estimate the infection risk of airborne indoor virus-transmissions, either calculation models or measurements can be carried out. The implemented simplifications of these calculation models (e.g., ideal mixed ventilation) might deliver fast but inaccurate results for certain scenarios. Previous experimental studies examine the effects of ventilation measures in more detail. However, instead of transient and absolute considerations of viral loads, they often regard relative statements of the infection risk.

Therefore, this study presents two experimental methods that are capable of determining a temporally and spatially resolved infection risk (absolute values) for different ventilation measures. Since particles entering from the outside would falsify the particle measurements, only the trace gas method is suitable for ventilation systems with air exchange and for natural ventilation. However, an accurate assessment of air purifiers based on filtration is only applicable by the surrogate particle method because trace gas is not filtered and therefore no device effect can be measured. For this reason, two different methods are essential.

Both methods are based on the theory that particles of relevant scales for infection procedures are airborne. The release of a controlled rate of either trace gas with a mass flow controller or particles with an aerosol generator allows a simulation of an infectious person releasing virus material. The measurement equipment includes an infrared spectrometer (trace gas method) or optical particle counters (surrogate particle method). For both approaches, the mathematical transfer of measured concentrations into infection risks is presented. In order to prove that the two methods are concordant, a comparison is essential. In an air visualisation laboratory with filtered outside air exchange, both methods are executed simultaneously. In fact, they provide similar results. This allows a first-time reliable experimental comparison of ventilation systems, natural ventilation and air purifiers.

Besides the detailed explanations and the comparison of both methods, several aspects that might influence the accuracy are discussed. Two exemplary scenarios show how practical both methods are and how scalable this principle is. Even if the ventilation concept deviates significantly from mixed ventilation, infection risks can be determined. Besides a two-person office, results of measurements performed in an exemplary classroom are presented. Both scenarios highlight the value of experimental investigations with temporal and spatial resolutions for determining infection risks. This allows, even for complex geometries, an assessment of the exposition time in rooms (e.g., workplace, events) and the identification of critical zones. On the one hand, the selection of a device and the related operating parameters such as volume flow can be determined. On the other hand, optimizations can be carried out (e.g., device positioning, orientation of supply air diffusers, permissible occupancy density). Furthermore, an evaluation of air purifiers beyond the quantity (Clean Air Delivery Rate, CADR) could be supplemented with information about locally resolved substance removal even under transient conditions.

## Figures and Tables

**Figure 1 ijerph-19-11300-f001:**
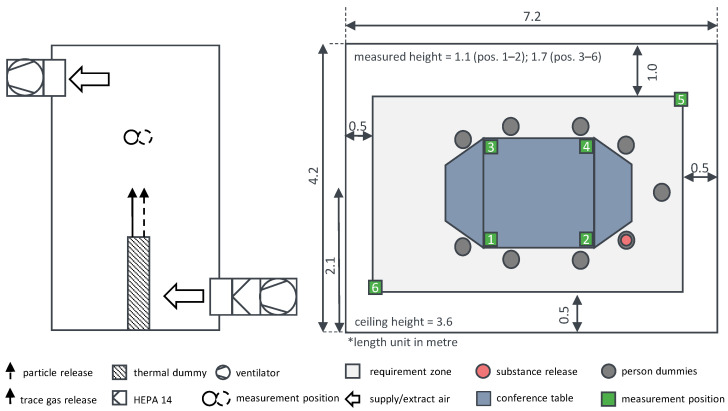
Set up to compare the two methods (**left**) and top view of the conference room (**right**).

**Figure 2 ijerph-19-11300-f002:**
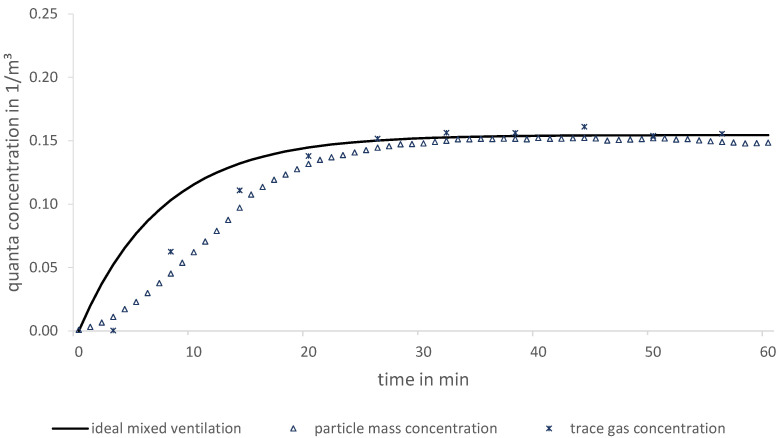
Comparison between particle and trace gas measurement as a function of quanta concentration over time exemplified in a conference room (position 3).

**Figure 3 ijerph-19-11300-f003:**
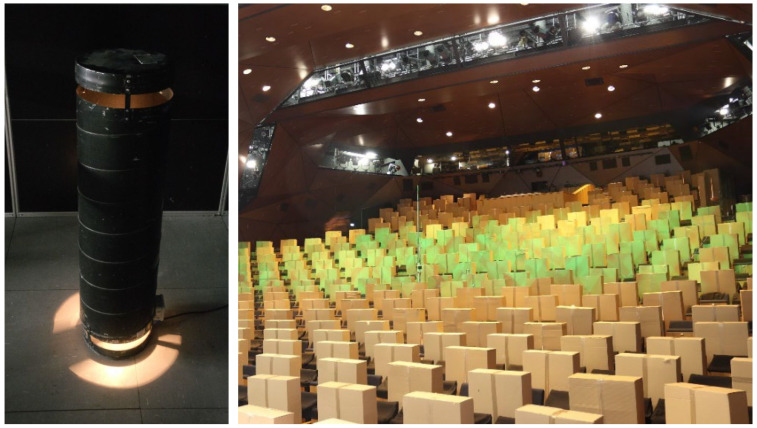
Professional thermal person dummy (**left**), low cost variants in Stuttgart Drama Theatre (**right**).

**Figure 4 ijerph-19-11300-f004:**
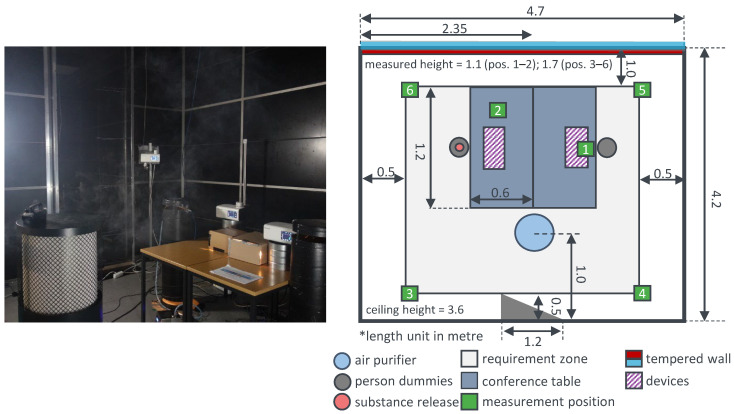
Photo (**left**) and the top view (**right**) of the experimental set up of a two-person office with air purifiers in an air visualisation laboratory. * length unit in metre.

**Figure 5 ijerph-19-11300-f005:**
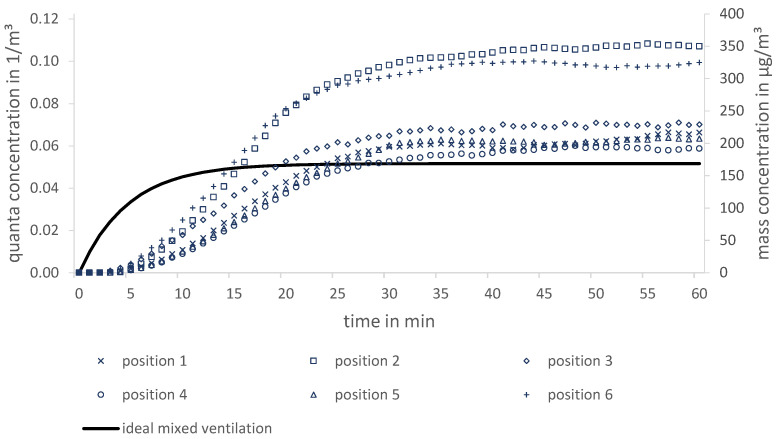
Theoretical (ideal mixed air) and measured quanta concentration (left axis) and their related mass concentration (right axis) of six positions in a two-person office.

**Figure 6 ijerph-19-11300-f006:**
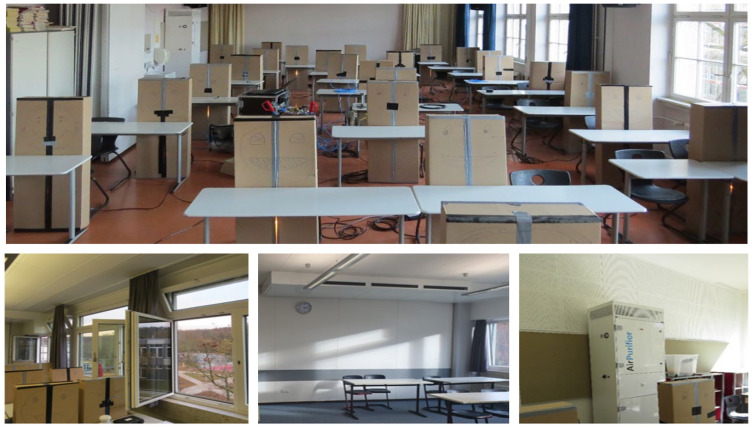
Experimental set up (**top**), exemplary window ventilation (**bottom left**), exemplary decentralised ventilation system (**bottom middle**), exemplary air purifier (**bottom right**).

**Figure 7 ijerph-19-11300-f007:**
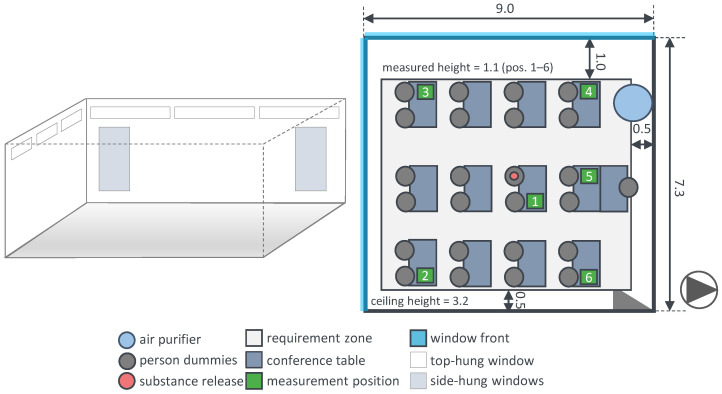
Draft (**left**) and the top view (**right**) of the experimental set up of a classroom with an air purifier.

**Figure 8 ijerph-19-11300-f008:**
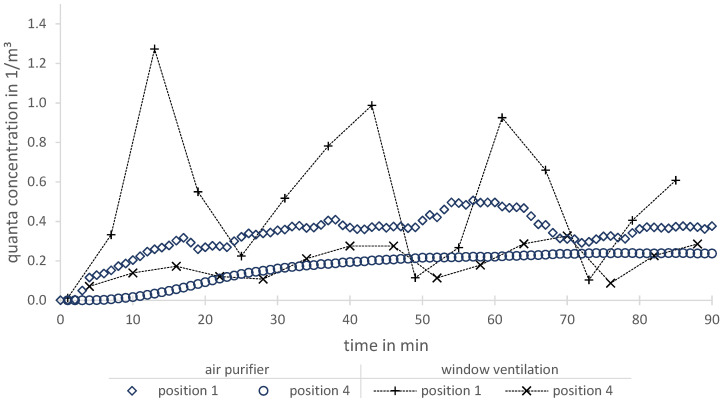
Comparison of quanta concentrations for a window ventilation and an air purification in a classroom.

**Figure 9 ijerph-19-11300-f009:**
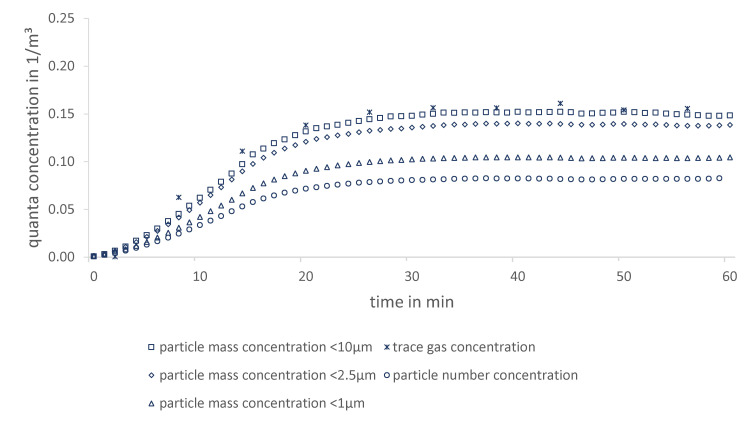
Comparison of quanta concentrations based on particle mass, particle number and trace gas concentrations in a conference room.

**Table 1 ijerph-19-11300-t001:** Relevant parameters of the measurement devices.

Surrogate Particles		Trace Gas	
Quantity	Value	Quantity	Value
medium	DEHS	medium	SF_6_
aerosol generator	Palas PAG 1000	MFC	Bronkhorst F-201CV
aerosol volume flow	168 L/h	trace gas volume flow	0.5 L/h
OPC	Palas AQ Guard	FTIR	Gasmet DX4015

**Table 2 ijerph-19-11300-t002:** Relevant parameters of the comparative measurement.

Quantity	Value	Quantity	Value
room volume	109 $m^{3}$	ventilation volume flow	900 $m^{3}$$h^{-1}$
quanta rate (output)	46.5 $h^{-1}$	exposition time	1 h
medium	SF_6_/DEHS	number of persons	9

**Table 3 ijerph-19-11300-t003:** Relevant parameters of the experiments concerning the effectiveness of air purifiers.

Quantity	Value	Quantity	Value
room volume	71 $m^{3}$	volume flow	900 $m^{3}$$h^{-1}$
quanta rate (output)	46.5 $h^{-1}$	exposition time	1 h
medium	DEHS	filtration class	HEPA 14
number of persons	2		

**Table 4 ijerph-19-11300-t004:** Boundary conditions of the experiments in the classroom.

Quantity	Value	Quantity	Value
**general parameters**			
room volume	210 $m^{3}$	number of persons	25
quanta rate (output)	46.5 $h^{-1}$	exposition time	1.5 h
medium	SF_6_/DEHS		
** $\mathrm{weather}\mathrm{conditions}{\;}^{1}$ **			
air temperature (outside)	14 ${}^{\circ }C$	air temperature (inside)	22 ${}^{\circ }C$
wind velocity	1.5 m $s^{-1}$	wind direction	north north east
**air purifier**			
volume flow	630 m^3^ $h^{-1}$	filter class	ePM1 85%/H13
**west-facing windows (5×)**		**south-facing windows (3×)**	
$\mathrm{window}\mathrm{type}{\;}^{2}$	th (3×) ∣ sh (2×)	$\mathrm{window}\mathrm{type}{\;}^{1}$	th (3×)
$\mathrm{window}\mathrm{area}{\;}^{3}\left(\mathrm{total}\right)$	4.5 $m^{2}$∣3.2 $m^{2}$	$\mathrm{window}\mathrm{area}{\;}^{3}\left(\mathrm{total}\right)$	4.5 $m^{2}$
tilt angle	17${}^{\circ }$ ∣ 90 ${}^{\circ }$	tilt angle	17${}^{\circ }$

^1^ mean values. ^2^ th: top-hung, sh: side-hung. ^3^ clear opening.

**Table 5 ijerph-19-11300-t005:** Curve agreement evaluation of different approaches.

Approach	Curve Agreement Evaluation Ψ
trace gas (reference)	0%
particle number concentration	48.7%
particle mass concentration <1 μm	35.0%
particle mass concentration <2.5 μm	13.3%
particle mass concentration <10 μm	5.6%

## Data Availability

The datasets generated/analysed during the current study are available from the corresponding author on reasonable request.

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
