# Peer review of "Experimental Methods of Investigating Airborne Indoor Virus-Transmissions Adapted to Several Ventilation Measures"

_ijerph, 2022, doi:10.3390/ijerph191811300_

Round 1
Reviewer 1 Report (Previous Reviewer 2)
Dear authors,
Once again, the paper has been uploaded without using the journals manuscript format. Please use the format which can be very easily found on the journals web page before submitting. I do not know why you are using the “springer nature” format when you are submitting to a different journal.
Introduction – add citations to prove your statements – here are some examples:
· first statement: “The ongoing pandemic teaches us that the implementation of appropriate
ventilation measures can significantly reduce indoor infection risks.” Add citations
· “In practice, different ventilation principles – natural ventilation and mechanical ventilation (mixing ventilation, displacement ventilation, downward ventilation) – are applied, depending on the type of use and the occupancy density of the room.” – add citation
For the experimental set up (fig 1) – the introduction to the experiments is still unclear and I do not know if this is done buy the authors or was done by someone else. Same comment for the experiment in point 5
Author Response
Dear Reviewer,
Thank you for the positive feedback on our manuscript. We have now transferred the manuscript into the MDPI template and incorporated your change requests directly into the PDF and marked them accordingly with a comment function. In some cases, we have highlighted additional text passages with a blue font colour. We hope that the measures taken are in line with your change requests.
Best regards
Lukas Siebler, Maurizio Calandri, Torben Rathje and Konstantinos Stergiaropoulos

Reviewer 2 Report (Previous Reviewer 3)
The manuscript combines experiments and calculations for various room types and ventilation methods for assessment of virus transmission risk. The presentation is clear, easy to follow for the reader. The manuscript is suitable for publication.
Author Response
Dear Reviewer,
Thank you for the very positive feedback on our manuscript. We have now transferred the manuscript to the MDPI template and incorporated the few change requests of the other revision directly into the PDF and marked them accordingly with a comment function. In some cases, we have highlighted additional text passages with a blue font colour.
Best regards
Lukas Siebler, Maurizio Calandri, Torben Rathje and Konstantinos Stergiaropoulos

Round 2
Reviewer 1 Report (Previous Reviewer 2)
Thank you for taking into account my remarks.
This manuscript is a resubmission of an earlier submission. The following is a list of the peer review reports and author responses from that submission.
Round 1
Reviewer 1 Report
In this manuscript tools to estimate the indoor infection risk are presented. A theoretical approach, assuming homogeneous distribution in the whole room, is shown. In addition, results from experiments, placing one ‘virus source’ and several sensors in a room, to determine the spatial distribution are given.
To estimate infection risk and to compare measures to reduce the risk are very important, this manuscript may help on this way
However, a lot remains unclear. To my opinion, it should be considerably improved by a revision
In particular, the description od the experiment and also of the data evaluation is very short and incomplete.
Nothing is said about the instruments used (how are the DEHS particles distributed in the room, what type of OPC’s are used, what is the concentration and size distribution of the dispersed particles. This information is needed to see, if agglomeration could be a problem.
The theory for evaluating gas of particles is the same, 3.1 and 3.2 contain the same information, can be combined.
P13, bottom: ‘With regard to suspected agglomeration effects a mass range up to 10 μm’, hard to understand. I guess this means particles up to a diameter (which diameter, optical, aerodynamic…?) should be considered for the evaluation of the mass. If the authors would show a size distribution of the particles it would be obvious, which sizer range has to be considered.
Most ambient particles are in the size range below one Micrometer, producing larger test particle could help to distinguish between test- and ambient- particles, the overlap would be small. Another option could be to use ‘labeled’ particles, e.g. containing a fluorescent dye and using a detector, sensitive to the fluorescence. Keeping the room free of ambient particle can be done in a lab, but not in an office, classroom etc., therefore this separation is essential to make the technique really applicable.
Table 2 and 3: the chosen flow rate is extremely high, more that 10 air exchanges per hour, this is not realistic. In Fig. 6 for example a ‘tempered wall’ is shown, the effect of this wall will be much higher, if a realistic air exchange rate is chosen.
No results for the opera are shown, therefore it makes not much sense to mention it
P17: ‘Under reproducible boundary conditions in an air visualization laboratory, the different air purifiers have been investigated and assessed’ no results for different purifiers are shown
P19: ‘The volume flow rates stated by the manufacturers of air purifiers (used for this calculation) seem to be less accurate than own measurements of a mechanical ventilation in an air duct (see fig. 4) ’ this is a very qualitative statement, quantify
P20: ‘20/5/20: 20 min opened, 5 min closed’ is this really correct, very unusual window ventilation with the window open almost all the time.
Fig,9,10: very little information is given, what is the spatial distribution in the classroom with window ventilation and air purifier, more information would help to draw conclusions from these results
Fig.11: knowing the size distribution of the particles would help to understand this figure and the results in Table 5 and if agglomeration ev. can be a problem, therefore it is essential to show the number concentration
P23: ‘To avoid underestimated infection risks it is highly recommended to apply mass concentrations < 10 μm’ what does this mean???
P24: have any SMPS measurements been done?
‘It is worth mentioning that with this approach agglomeration effects can not be detected appropriately and therefore results in inaccuracies’. As mentioned above, the effect of agglomeration can easily estimated from the number concentration.
It would be nice to get some conclusions, how can these results be used to improve indoor air quality, compare effects of ventilation, local air purifiers, flow strategy etc.
Author Response
Dear Reviewer,
Thank you very much for your valuable comments and remarks. We have attached a PDF. In addition to the list of general adjustments and improvements to the script, it also contains detailed replies to all comments and below the entire revised script.
We hope that the adjustments meet your scientific expectations and look forward to receiving further feedback.
Kind regards
Lukas Siebler, Maurizio Calandri, Torben Rathje, Konstantinos Stergiaropoulos

Reviewer 2 Report
Dear Author(s),
Your article is interesting however it is very chaotic and difficult to connect the chapters together. Bellow you can find my remarks:
There are no line numbers so I will try to point out the places that need to be changed using some of the text.
1. Introduction:
a. “Using idealised assumptions…..”, “Displacement ventilation concepts…..”, “In
reality, however….” – add literature to each example to prove this is backed by literature.
b. You discuss in the abstract about natural ventilation, please add literature to the introduction about improvement of contaminant removal using natural ventilation and mixed ventilation methods in different airtight structures like:
i. Szczepanik-ÅšcisÅ‚o, Nina “Air leakage modelling and its influence on the air quality inside a garage”, E3S Web of Conferences, Volume 443, July 2018
ii. Schnotale, J. “CFD simulations and measurements of carbon dioxide transport in a passive house” Refrigeration Science and Technology, Pages 4065 – 4072, 2015, 24th IIR International Congress of Refrigeration, ICR 2015 Yokohama
2. In the chapters: Theory of airborne virus infections, Continuous ventilation and Periodic ventilation – please give explanation how this is involved in the study. While I understand it will be used for the analysis, please give more detail how these factors can be used. If any of them are not used, then remove them form the article.
3. The same comment as point 2 for the chapters about the tracer gas
In general, the article is hard to follow as you put in a lot of theory and a lot of equations and then do not explain what you are using them for and how they are relevant. Either divide the article into the elements needed for CFD and then experimental studies or remove the information that is not necessary. And add a better description in the introduction and abstract of what you are doing as there is a lot of information in the article, but it is not clear what goes with what.
4. In the chapters Experimental studies – for the examples please add when and by who they were conducted and if the research has been published then add citation. It is not clear if this is your research or someone else’s as they are shown quickly and with little detail.
Author Response

(The authors gave the same response as above.)

Reviewer 3 Report
The manuscript combines experiments and calculations for various room types and ventilation methods for assessment of virus transmission risk. The presentation is clear, easy to follow for the reader.
The manuscript needs minor revision.
Comments:
1. p.4 para 3, l.4: "in [8]" --> "in Li et al. [8]"
2. p.4 para 3, l.5: "simulations are used to simulate" awkward, should be reworded
3. p.5 last sentence: Reference missing for "Wells et al."; "Riley et al." --> "Riley et al. [11]"
4. p.14 l.1, l.3: "between both" --> "between the two" in title of Section 4 and following text
5. p.14 l.4,8: reference number in brackets should be moved immediately after "Bivolarova et al."
6. p.15, caption of Fig.3: "compare both" --> "compare the two"
Author Response

(The authors gave the same response as above.)

Round 2
Reviewer 2 Report
Dear Authors, the article has not been updated according to my remarks. Additionally, the editing of the article is not according to the journal guidelines – this is extremely unprofessional and should not happen.
The changes in the manuscriptal are not marked so I cannot verify properly what has been changed.
The chapter 5 you say, “In the following section it is shown that both methods are practical.” Yet you do not compare them to the theoretical background.
I still do not know if the experiments in the study were done by your team or someone else. They are poorly described compared to the mathematical background